# FEEDBACK PROMOTES EFFICIENT-CODING WHILE AT-TENUATING BIAS IN RECURRENT NEURAL NETWORKS

## ABSTRACT

Studies of human decision-making demonstrate that environmental regularities, such as natural image statistics or intentionally nonuniform stimulus probabilities, can be exploited to improve efficiency (termed 'efficient-coding'). Conversely, from a machine learning perspective, such nonuniform stimulus properties can lead to biased neural networks with poor generalization performance. Understanding how the brain flexibly leverages stimulus bias while maintaining robust generalization could lead to novel architectures that adaptively exploit environmental structure without sacrificing performance on out-of-distribution data. To address this disconnect, we investigated the impact of stimulus regularities in a 3-layer hierarchical continuous-time recurrent neural network (ctRNN) to better understand how artificial networks might exploit statistical regularities to improve efficiency while avoiding undesirable biases. We trained the model to reproduce one of six possible inputs under biased conditions (stimulus 1 more probable than stimuli 2-6) or unbiased conditions (all stimuli equally likely). Across all hidden layers, more information was encoded about high-probability stimuli, consistent with the efficient-coding framework. Importantly, reducing feedback from the final hidden layer of trained models selectively magnified representations of high-probability stimuli, at the expense of low-probability stimuli, across all layers. Together, these results suggest that models exploit nonuniform input statistics to improve efficiency, and that feedback pathways evolve to protect the processing of low-probability stimuli by regulating the impact of biased input statistics.

## 1 INTRODUCTION

Neural systems have limited capacity to encode sensory inputs (Barlow, 1961; Burgess et al., 1981) and are thought to allocate resources to match the statistics of natural environments (Simoncelli & Olshausen, 2001; Geisler, 2008). In line with this notion of efficient-coding, recent human studies suggest that expectations about stimuli that are most likely to occur in a given context can similarly improve information processing and lead to faster, more accurate decisions. However, the nature of these expectation effects is poorly understood, and it is unclear whether they arise from bottom-up changes within early sensory areas or if they are generated by top-down signals that originate in higher-order brain regions (Summerfield & De Lange, 2014; Rungratsameetaweemana et al., 2018; Harrison et al., 2023). For example, when searching for your red key chain, expectations about the most likely location could enhance responses of sensory neurons tuned to red objects. Alternatively, expectations could lower the decision threshold, thereby speeding choices. Dissociating these mechanisms in human experiments has proven difficult, in part because directly measuring top-down feedback and decision thresholds is challenging, particularly in non-trivial tasks (Rungratsameetaweemana & Serences, 2019; De Lange et al., 2018). Computational modeling provides a principled way to disentangle these processes, enabling expectation effects to be experimentally isolated from other top-down factors and supporting the ability to pinpoint where in the processing hierarchy expectations influence information processing.

In many ways, AI research parallels work in human neuroscience, particularly in the domains of efficient-coding and expectations. For example, techniques such as pruning and regularization are used to eliminate redundant information while retaining performance, and reusing computational motifs that can be shared across tasks improves efficiency and supports learning with minimal cross-task interference (Hoefler et al., 2021; Duncker et al., 2020; Driscoll et al., 2024). More-

over, much research on continuous time series classification - including Transformer-based models, CNNs, and LSTMs - has focused on developing training strategies that avoid overfitting and unnecessary network growth that can arise from using natural training data (Sun et al., 2023; Lui et al., 2024). Finally, it is well documented that training models with unbalanced data can amplify existing biases (Buda et al., 2018; He & Garcia, 2009; Mazurowski et al., 2008; Japkowicz & Stephen, 2002), consistent with work showing that expectations about stimulus probabilities can bias human decision making (Summerfield & De Lange, 2014; Bang & Rahnev, 2017). However, exploiting inhomogeneities in training data can be beneficial in cases where similar inhomogeneities exist at test time, so failing to exploit regularities can sometimes be disadvantageous. Thus, theoretical work that characterizes expectation-related biases might reveal insights about how to combat overinflated bias in artificial neural networks while still preserving the benefits associated with efficiently coding more probable stimuli.

In this work, we examine the structural locus of expectation-related biases using hierarchical continuous-time recurrent neural networks (ctRNNs) trained with statistically nonuniform input stimuli. Using a biologically plausible hierarchical system modeled after the human brain, we provide a framework to investigate how bottom-up sensory signals and top-down expectation signals interact to shape representations and influence decisions. Using a 3-layer ctRNN, we first show that more information about expected stimuli is encoded, compared to unexpected, in all hidden layers. This result demonstrates that the models learned to prioritize and efficiently process stimuli that were more probable during training. We then show that reducing the strength of feedback from the last hidden layer magnified expectation-based biases in all hidden layers, particularly when information about the stimulus-to-output mapping was not provided until after stimulus offset. This latter finding reveals that feedback connections evolved during training to attenuate the over expression of biases so that reasonable performance levels were still preserved for unexpected stimuli.

## 2 RELATED WORK

**Neuroscience and Behavior** Nonuniform stimulus statistics and expectations about likely context-dependent stimuli facilitate the speed and accuracy of human decisions. For example, the *oblique effect* refers to an enhanced ability to discriminate small orientation changes for vertical and horizontal (cardinal) stimuli compared to diagonal (oblique) stimuli (Appelle, 1972). Consistent with this effect, there are more orientation-selective neurons in visual cortex with a preference for cardinal orientations (Li et al., 2003; Kreile et al., 2011; Appelle, 1972). Biases in behavioral performance and neural tuning are thought to be driven by the principle of efficient-coding, where limited neural resources are allocated to match the overrepresentation of cardinal orientations in natural images (Barlow, 1961; Girshick et al., 2011). Indeed, during development (i.e. 'training'), overexposure to specific features can shift the distribution of neural tuning in visual cortex in line with this principle (Blakemore & Cooper, 1970; Kreile et al., 2011).

In addition to exploiting irregularities in natural stimulus statistics, context-dependent expectations can increase the precision of feature-selective activation patterns in early visual cortex to favor the processing of likely over unlikely stimuli (Kok et al., 2012; 2013). However, such modulations may reflect co-occurring changes in stimulus relevance (attention) rather than expectations (Rungratsameetaweemana & Serences, 2019). EEG markers of neural processing suggest that context-dependent expectation can influence later stages of processing, consistent with lower decision thresholds or top-down feedback (Rungratsameetaweemana et al., 2018). This mix of findings was a primary motivation for studying the multi-layer ctRNNs to assess the role of feedback in mediating expectation-induced bias.

**Machine Learning** Analogous to the neuroscience literature, DNNs trained on natural images exhibit an oblique effect such that more neurons respond to cardinal orientations, and these neurons have higher Fisher Information, suggesting that networks exploit natural image statistics to encode images with higher fidelity (Henderson & Serences, 2021). The adoption of efficient-coding strategies when trained on natural images corresponds to abundant observations that models trained on any type of biased or unbalanced data will reflect and even amplify these biases (Buda et al., 2018; He & Garcia, 2009; Mazurowski et al., 2008; Japkowicz & Stephen, 2002). Various approaches from data-level (Chawla et al., 2002; Van Hulse et al., 2007; He et al., 2008; Buda et al., 2018) and algorithm-level methods (Zhang & Wang, 2013; Khan et al., 2018; Johnson & Khoshgftaar, 2019)

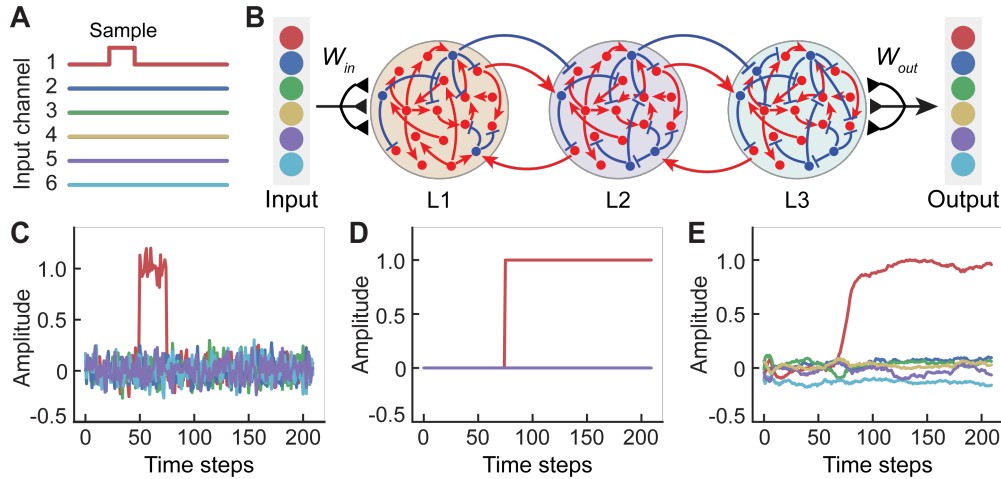

Figure 1: (A) Task stimulus space. (B) ctRNN architecture. (C) Example input trial. (D) Target model output. (E) Example correct model output.

have been applied to eliminate performance biases resulting from unbalanced datasets. However, as long appreciated in the neuroscience literature, some biases can be adaptive to support faster and more accurate information processing, particularly if the biases in the training data are known to generalize out-of-distribution. Thus, we intentionally trained ctRNNs with an imbalanced dataset to probe how learned biases impact different stages of the information processing hierarchy. We focus here on causal analyses via targeted ablations to neural networks to discover which structural components are necessary for performance (Meyes et al., 2019), with an emphasis on examining the role of feedback connections in mediating expectation-related biases. Conceptually, our study is related to Henderson & Serences (2021), which demonstrated that DNNs naturally adopt efficient-coding strategies during training. However, Henderson & Serences (2021) used time-invariant feedforward networks that lacked biological plausibility and thus did not provide the opportunity to examine the influence of top-down feedback on regulating expectation-induced biases during training.

## 3 CONTINUOUS-TIME RECURRENT NEURAL NETWORK

### 3.1 METHODS

#### 3.1.1 MODEL ARCHITECTURE

We constructed continuous-time recurrent neural networks (ctRNNs) (Song et al., 2016; Kim et al., 2019) with biologically inspired constraints. The network consisted of an input layer with one channel per stimulus, three recurrent hidden layers, and an output layer with one channel per stimulus (Fig. 1A). The three sequentially connected hidden layers each contained $N$ units ($N = 200$) and the first hidden layer received input $I_{ext}$ (Fig. 1B). The activity of each unit in each hidden layer is governed by the following set of equations:

$$\tau_i^{(1)} \frac{dx_i^{(1)}}{dt} = -x_i^{(1)} + \sum_{j=1}^{N} w_{ij}^{(1)} r_j^{(1)} + I_{ext}$$

$$\tau_i^{(2)} \frac{dx_i^{(2)}}{dt} = -x_i^{(2)} + \sum_{j=1}^{N} w_{ij}^{(2)} r_j^{(2)} + \sum_{j=1}^{N} w_{ij}^{(2,1)} r_j^{(1)} \quad (1)$$

$$\tau_i^{(3)} \frac{dx_i^{(3)}}{dt} = -x_i^{(3)} + \sum_{j=1}^{N} w_{ij}^{(3)} r_j^{(3)} + \sum_{j=1}^{N} w_{ij}^{(3,2)} r_j^{(2)}$$

where $\tau_i^{(k)}$ is the synaptic decay time constant of unit $i$ in layer $k$, $x_i^{(k)}$ is the synaptic input current of unit $i$ in layer $k$, $r_i^{(k)}$ is the firing rate of unit $i$ in layer $k$, $w_{ij}^{(k)}$ is the synaptic connection weight

from unit $j$ to unit $i$ in layer $k$, $w_{ij}^{(k,l)}$ is the synaptic connection weight from unit $j$ in layer $l$ to unit $i$ in layer $k$, and $I_{ext}$ is the external input. Connections in the hidden layers were strictly excitatory or inhibitory (Dale's principle) where the probability of a connection between any units $i$ and $j$ within a layer was set to 0.2. If connections were formed between two units, 0.8 were excitatory and 0.2 inhibitory (Kim et al., 2017; Wei et al., 2022). As shown in the schematic diagram of the model (Fig. 1B), the three hidden layers are sequentially connected via feedforward and feedback connections, with only excitatory units relaying inter-layer signals (Melzer & Monyer, 2020).

We estimated the firing rate of unit $i$ ($r_i$) by passing the synaptic current variable ($x_i$) to a sigmoid function:

$$r_i = \phi(x_i) = \frac{1}{1 + \exp(-x_i)}$$

The model received time-varying input signals along with a Gaussian white noise variable:

$$I_{ext} = w_{in} \cdot u + \mathcal{N}(0, 0.01)$$

where the task-specific stimulus signals ($u$) were given to the model via a set of weights ($w_{in}$), and $\mathcal{N}(0, 0.01)$ represented a random Gaussian noise with zero mean and variance of 0.1. As shown in Fig. 1C.

The output of the multi-layer model at time $t$, ($o(t)$) was computed as a linear readout of the population activity of the last hidden layer:

$$o(t) = w_{out} \cdot r^{(3)}(t)$$

where $w_{out}$ refers to the weights in the output layer.

### 3.1.2 TASK AND TRAINING

The network was trained on a probabilistic associative learning task where one input stimulus out of six possible stimuli was presented to the input layer more often than the rest of the stimuli during training. Specifically, the 'expected' stimulus was presented on 70% of the training trials, while the other five 'unexpected' stimuli were each presented on 6% of the training trials. The stimulus signals were modeled as white-noise signals (drawn from the standard normal distribution) with a constant offset value of 1.0 added during the stimulus window (Fig. 1C). The network model was trained to produce an output signal approaching +1 in the corresponding output channel and 0 in all other channels (Fig. 1D). For comparison, we also trained another group of models where all stimuli were equally represented in the training set. Twenty models were trained for each training condition. Moving forward, we refer to models trained with nonuniform stimulus probabilities as "biased models" and those trained with uniform stimulus probabilities as "unbiased models".

Model parameters ($\tau$, $w$, $w_{out}$) were optimized using backpropagation through time (BPTT) with the Adam optimizer (Kingma & Ba, 2017) (learning rate = 0.01, batch size = 256), minimizing mean squared error between the target output (Fig. 1D) and the actual output (Fig. 1E). Model training continued until reaching a loss <0.001 or an accuracy of >90%. All models reached the accuracy threshold of >90% before reaching the loss cutoff.

### 3.1.3 EVALUATION

After training, we evaluated 20 independently initialized models on a new balanced dataset of 2,400 trials. Importantly, all evaluation trials for biased and unbiased models were generated using a uniform stimulus distribution so that all 6 stimuli were equally probable. We also manipulated the amount of noise in the evaluation stimuli: the level used during training $\mathcal{N}(0, 0.01)$ and a higher level $\mathcal{N}(0, 0.06)$. This noise manipulation was used to determine if any expectation effects were amplified under high-noise conditions. Based on the resulting activity in the three ctRNN hidden layers in each model, we assessed four metrics: (1) ctRNN evaluation accuracy, reflecting how often the correct stimulus was identified; (2) decoding accuracy of a classifier trained on trial-wise activity from each of the three ctRNN layers, estimating the robustness of stimulus representations; (3) the bias index ($\Delta$AUC), defined as the difference between decoding accuracy for the expected stimulus and the average decoding accuracy for all unexpected stimuli; and (4) neural trajectories of stimulus representations identified with principal components analysis (PCA). For detailed methods of statistical comparisons see Appendices B.2.

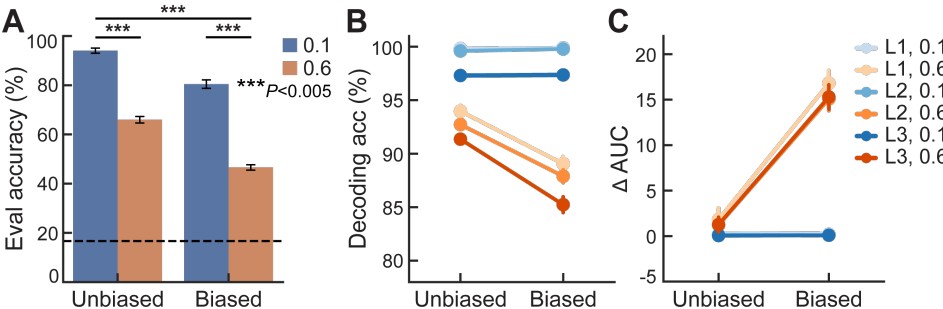

Figure 2: Colors: orange for high (0.6) and blue for low (0.1) noise added to stimuli. Color saturation: light is layer 1, medium is layer 2, dark is layer 3. All error bars are standard error. (A) ctRNN evaluation accuracy on balanced test trials. (B) Average decoding accuracy over time. (C) Expectation bias ($\Delta$AUC), the difference in decoding accuracy between the expected and unexpected stimuli.

### 3.2 CTRNNs RELY ON LEARNED BIASES WHEN STIMULI ARE NOISY

Increasing stimulus noise significantly reduced task accuracy in both biased and unbiased networks (Fig. 2A) ($P < 0.001$). Biased networks performed above chance but with lower overall accuracy compared to unbiased networks (biased, $0.64 \pm 0.20$ (mean $\pm$ stdev); unbiased, $0.80 \pm 0.17$ (mean $\pm$ stdev), $P < 0.001$). However, the magnitude of performance decline under noise did not differ significantly between biased and unbiased models (biased, $-0.34 \pm 0.16$ (mean $\pm$ stdev); unbiased, $-0.28 \pm 0.13$ (mean $\pm$ stdev); $P = 0.19$), indicating that biased networks were not differently impacted by stimulus noise. Errors in biased networks occurred primarily on trials with unexpected stimuli.

While behavioral accuracy decreased similarly across biased and unbiased models, decoding accuracy did not follow the same pattern. Instead, overall decoding accuracy, averaged over all time-points following stimulus onset, declined with increased noise ($P < 0.001$), with the largest reduction observed in the biased networks ($P < 0.001$) and with progressively larger effects of noise across higher layers of the network ($P < 0.001$; Fig. 2B). In contrast, without noise, decoding accuracy was comparable across layers, with only a slight decrease in layer 3.

Next, we examined expectation bias as quantified by $\Delta$AUC. In biased ctRNNs, bias became apparent when stimuli were evaluated under noisy conditions. When the noise level was kept low, biased ctRNNs exhibited only slight bias (Fig. 2C; $0.13 \pm 0.41$, mean $\pm$ stdev). However, when the noise level was increased, expectation bias was strongly revealed in the biased networks (Fig. 2C; $15.7 \pm 5.92$, mean $\pm$ stdev). Bias was slightly greater in layer 1 compared to layers 2 and 3 ($P = 0.006$). In contrast, unbiased networks showed no preference for stimulus 1 irrespective of stimulus noise, as reflected by $\Delta$AUC values near zero (Fig. 2C; $0.78 \pm 3.33$, mean $\pm$ stdev). Therefore, while both biased and unbiased ctRNNs experienced similar performance degradations with noise, decoding analyses uncovered a stronger representation of the expected stimulus in the biased networks.

### 3.3 FEEDBACK ABLATION LEADS TO INCREASED BIAS IN LAYER 3

We selectively ablated feedback connections to further examine biased representations across the network hierarchy. In our ctRNNs, there are only feedback connections from layers 2 to 1 and layers 3 to 2, with no skip connections directly from layers 3 to 1. We further imposed a constraint where only excitatory units provided feedback, to reflect the current lack of evidence for long-range inhibitory feedback connections in the brain (Melzer & Monyer, 2020). The following analyses compared the bias index when feedback strength was reduced to 0.7 (i.e., 70% of the training level of 1.0). We focused on this level of feedback reduction for clarity, but similar reductions in model performance and decoding accuracy were observed with further decreases in feedback strength, see Appendix Fig. C.1.

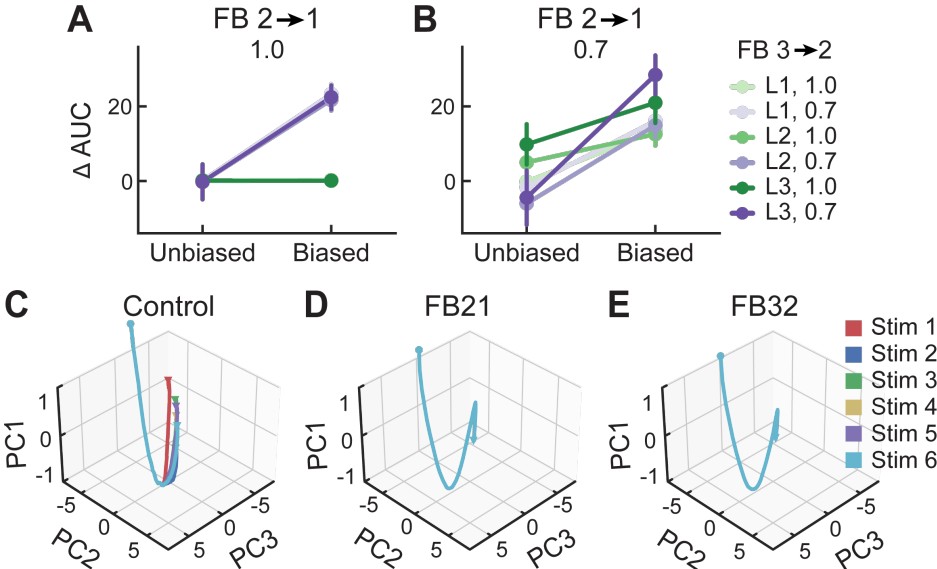

Figure 3: (A-B) Colors: purple for reduction (0.7) and green for full (1.0) feedback from layer 3 to 2. Color saturation: light is layer 1, medium is layer 2, dark is layer 3. Error bars are standard error. (A) Bias ($\Delta$AUC) in ctRNNs tested with full feedback from layer 2 to 1. (B) Bias in ctRNNs tested with reduced feedback from layer 2 to 1. (C-E) Circle is start and triangle is end of trajectory. (C) PCA trajectories of neural activity for each stimulus with full feedback strength. (D) Same as (C) with reduced feedback from layer 2 to 1 only. (E) Same as (C) with reduced feedback from layer 3 to 2 only.

Reducing feedback earlier in the hierarchy between layer 2 and 1 (FB21) and reducing feedback later in the hierarchy between layer 3 and 2 (FB32) both caused model evaluation accuracy to drop. Model accuracy for biased ctRNNs dropped to chance with FB32 reduction only (Appendix Fig. C.2A) and below chance with reductions in both FB21 and FB32 (Appendix Fig. C.2B). In contrast, model accuracy for unbiased ctRNNs dropped below chance for all types of feedback reductions (Appendix Fig. C.2A-B). Similarly, decoding accuracy decreased with a reduction in the feedback strength, but less so with biased ctRNNs (Appendix Fig. C.2C-D). Despite this drop, decoding accuracy remained well above chance with feedback ablations ($0.82 \pm 0.08$ (mean $\pm$ stdev); Appendix Fig. C.2C-D) and computing the bias metric ($\Delta$AUC) in each hidden layer revealed that bias increased along the hierarchy when feedback was reduced (Fig. 3A, B). We used PCA to visualize a low dimensional subspace representation of neural activity in all layers, and found that trajectories collapsed and failed to diverge into distinct representations when feedback was reduced (layer 3: Fig. 3D-E; all layers: Appendix Fig. C.4).

## 4 STIMULUS-OUTPUT CUE TO ENCOURAGE SUSTAINED STIMULUS REPRESENTATIONS

We hypothesized that with a simple one-to-one mapping task, the ctRNNs were able to learn the correct response without relying on hierarchical stimulus-response representations. In order to encourage separate representations of the stimulus and the corresponding response behavior, we modified the task by adding a stimulus-output mapping cue so that the model would have to sustain a representation of stimulus interdependently from the response mapping.

## 4.1 METHODS

### 4.1.1 MODEL ARCHITECTURE

The model architecture matched the previous ctRNNs with one change: to model the top-down nature of task instructions often employed in human studies, the last hidden layer received a cue $C$ indicating one of two possible stimulus-output mappings:

$$\tau_i^{(3)} \frac{dx_i^{(3)}}{dt} = -x_i^{(3)} + \sum_{j=1}^{N} w_{ij}^{(3)} r_j^{(3)} + \sum_{j=1}^{N} w_{ij}^{(3,2)} r_j^{(2)} + C$$

which is a time-varying input signal with Gaussian white noise:

$$C = w_{cue} \cdot c + \mathcal{N}(0,1)$$

where the stimulus-output mapping signals ($c$) were given to the model via a set of weights ($w_{cue}$) and $\mathcal{N}(0,1)$ represents a random Gaussian noise with zero mean and variance of 1. Each of the two possible stimulus-output mapping was pseudo-randomly generated for each model, with the constraint that each mapping was unique. For example, one model might map $\{1,2,3,4,5,6\} \mapsto \{4,1,3,2,6,5\}$, while another might map $\{1,2,3,4,5,6\} \mapsto \{1,4,5,2,3,6\}$.

### 4.1.2 TASK AND TRAINING

The task and training procedures were the same as in the previous ctRNNs, with the following exceptions. We manipulated cue onset time so that one set of models was trained with an early cue presented at trial onset, and another set of models was trained with a late cue presented immediately after stimulus offset. With an early cue, the models could almost immediately map the stimulus to a representation of the corresponding response. However, delaying cue onset would force the models to learn a sustained representation of the stimulus that was independent of the output, as the correct output was not signaled until stimulus offset. The cue identity on each trial was randomly selected with uniform probability during training and during evaluation.

### 4.1.3 EVALUATION

The evaluation procedure matched the previous ctRNN evaluation.

## 4.2 CUE TIMING AND TOP-DOWN BIAS

Cue timing did not affect overall ctRNN performance, with ctRNNs performing similarly to those trained without a cue (Fig. 4A). There were similar decreases in decoding accuracy with added stimulus noise as well (Fig. 4B-C). Decoding accuracy was lower with late (stimulus offset) cued ctRNNs when stimuli were noisy ($P = 0.003$) and this effect was greatest in layer 3 in biased ctRNNs ($P < 0.001$). Late cue models showed the same increased bias with stimulus noise as no-cue models. However, models trained with a late cue onset exhibited greater bias in layer 3 ($P < 0.001$; Fig. 4E). Thus, the performance of models trained with an early cue onset closely resembled the performance of models trained without a cue, and models with a late cue had magnified bias in layer 3.

## 4.3 FEEDBACK ABLATION REVEALS INCREASED BIAS IN LAYER 3

To further evaluate biased representations along the hierarchy of our networks, we conducted selective feedback ablation in ctRNNs trained and tested with different cue timing. Model performance at evaluation dropped as a result of feedback ablation in both FB21 and FB32. However, when the cue was later, both biased and unbiased models performed better than ctRNNs trained with an early cue when feedback was reduced from layer 3 to 2 (FB32) and when both FB32 and FB21 were reduced (Appendix Fig. C.3A-D). Decoding accuracy also decreased when feedback connections were ablated (Appendix Fig. C.3E-H) and later cue timing reduced decoding accuracy in biased ctRNNs such that they were similar to unbiased ctRNNs, whereas with the early cue timing decoding accuracy was higher in biased ctRNNs.

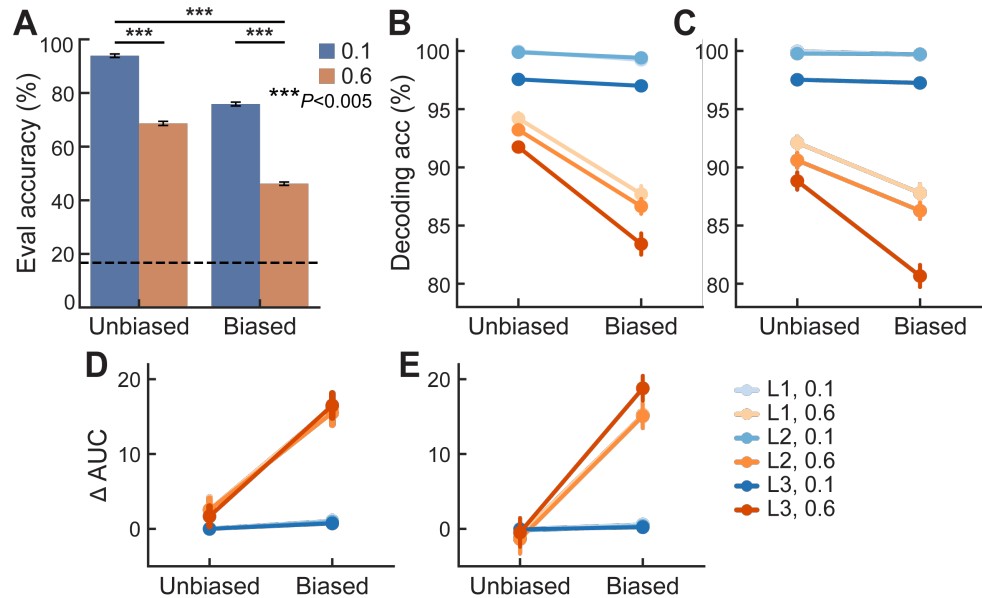

Figure 4: Colors: orange for high (0.6) and blue for low (0.1) noise stimuli. Color saturation: light is layer 1, medium is layer 2, dark is layer 3. Error bars are standard error. (A) ctRNN performance at evaluation, averaged over cue timing. (B) Decoding accuracy in ctRNNs tested with an early cue. (C) Same as (B) with late cue. (D) Bias ($\Delta$AUC) in ctRNNs tested with early cue. (E) Same as (D) with late cue.

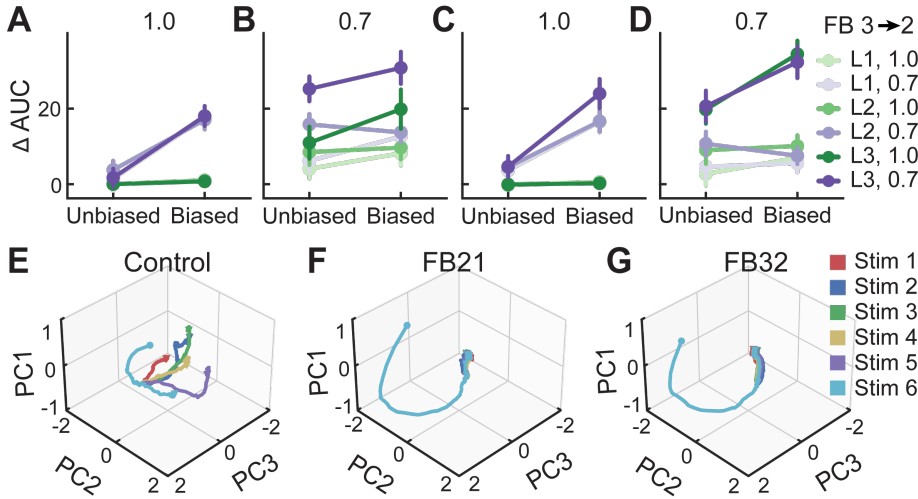

Figure 5: (A-D) Colors: purple for reduced (0.7) and green for full (1.0) feedback from layer 2 to 2 (FB32). Color saturation: light is layer 1, medium is layer 2, dark is layer 3. Error bars are standard error.(A) Bias ($\Delta$AUC) in ctRNNs with cue at start and full feedback from layer 2 to 1 (FB21). (B) Bias in ctRNNs with cue at start and reduced FB21. (C) Same as (A) with late cue. (D) Same as (B) with late cue. (E-G) Circle is start and triangle is end of trajectory. Example late cue ctRNN (for early cue see Appendix Fig.C.6). (E) PCA trajectories of neural activity for each stimulus with full feedback strength. (F) Same as (E) with reduced feedback from layer 2 to 1. (G) Same as (E) with reduced feedback from layer 3 to 2.

Feedback ablations between layers 2 and 1 and layers 3 and 2 revealed increases in bias (Fig. 5A-D), with bias showing the strongest effects in layer 3 for both cue onset times ($P < 0.001$). There was an interaction with cue timing and FB21 such that when FB21 was reduced and there was a late cue, bias

was high and did not change with the addition of reducing FB32 (Fig. 5D; $P < .05$). Additionally, late cues revealed an increase in bias in later layers with the reduction of FB32 (Fig. 5C; $P < 0.001$), whereas when the cue was early, layer 3 and 2 had similar levels of bias (Fig. 5A). Together, these results further highlight the importance of feedback from layer 3 in propagating bias signals. When feedback was reduced at any level, layer 1 neural activity did not produce biased decoding. Finally, principal components analysis of neural activity in layer 3 revealed a collapsed set of representations for each stimulus when feedback is reduced at any level in both late cue (layer 3: Fig. 5E-G; all layers: Appendix Fig. C.5) and early cue ctRNNs (all layers: Appendix Fig. C.6)). Collapsed trajectories along with decreased model performance indicate that feedback is essential to generate separable representations of both expected and unexpected stimuli.

## 5 Discussion

### 5.1 Conclusion

We used biologically inspired hierarchical ctRNNs to study the mechanisms that produce expectation-related biases that arise when models are trained using unbalanced training data. First, we found that feedback pathways are essential to support robust model performance, as attenuated feedback results in collapsed stimulus trajectories in low-dimensional subspaces. Second, we found that representations were more biased in later layers, supporting the hypothesis that expectation-related biases are more pronounced in higher-order brain areas more closely linked to decision-making. Third, ablating feedback across layers, particularly from the final layer, decreased model performance overall and led to an over expression of bias across all 3 hidden layers. Together, these findings suggest that feedback pathways acquire a compensatory role during training to attenuate prior-induced bias. In turn, this regulatory feedback supports efficient processing of expected inputs while also ensuring sufficiently robust processing of unexpected inputs.

### 5.2 The Role of Bias in efficient-coding: from Neuroscience to AI

Our results strengthen the evidence that top-down processes facilitate neural processing of high-probability stimuli. Importantly, our experiments revealed a nuanced relationship between biased training and biased behavior: while our ctRNNs were trained on unbalanced data, they exhibited minimal biased behavior unless perturbed with noise or feedback ablations. This suggests that hierarchical neural networks can harness statistical regularities in their training environment to develop efficient representations without producing biased outputs due to the compensatory influence of feedback pathways. This finding has important implications for both neuroscience and AI, especially since natural environments are filled with inherently unbalanced stimulus probabilities that can be exploited to improve efficiency without leading to the excessive expression of biases that overpower representations of less likely stimuli.

### 5.3 Limits and Future Directions

First, our architecture does not fully capture the hierarchical structure observed in human cortical circuits. In our design, each hidden layer has the same number of units and sparsity constraints. In contrast, empirical studies show systematic differences in the cortical regions we aim to model, with particularly high neuronal density in early sensory processing areas (Collins et al., 2010) and distinct characteristics of connectivity throughout the hierarchy (Felleman & Van Essen, 1991). Second, our study focused on a relatively simple reproduction task, leaving open questions about how biases operate in more naturalistic scenarios. Extending this framework to investigate cross-modal expectations in more complex tasks could provide deeper insights into general computational principles underlying bias and efficient-coding in more naturalistic settings.

### Reproducibility

To ensure reproducibility of our results, all code will be made available upon publication.

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

## APPENDICES

## A CTRNN DETAILS

We trained 20 ctRNNs for each task condition initialized from different random seeds (1–20). The seed values were reused across conditions (e.g., both the no-cue unbiased and no-cue biased conditions each had models with seeds 1–20). Thus, within each condition the models were unique, but the seed sets were parallel across conditions, enabling condition-to-condition comparability. Weights were initialized from a Gaussian distribution ($\mu = 0$, $\sigma = 1$) with layer-specific gain of 1.5, and the probability of recurrent connections between units in a layer was set to 20%, with 80% excitatory connections and 20% inhibitory connections. Dale's principle was enforced by fixing the sign of connections after initialization.

Table 1: Model and task parameters.

| Category | Parameter | Value |
|---|---|---|
| Task / Stimulus | Task type | No cue, Cue |
| | Trial length ($T$) | 210 time steps |
| | Stimulus onset / duration | 50 / 25 time steps |
| | Stimulus noise ($\sigma_{\text{stim}}$) | $\mathcal{N}(0, \sigma^2)$, $\sigma = 0.1$ |
| | Stimulus probability | biased: 0.7; unbiased: $1/6$ |
| Network | Hidden layers | 3 |
| | Units per layer | 200 |
| | Internal noise ($\sigma_{\text{int}}$) | $\mathcal{N}(0, \sigma^2)$, $\sigma = 0.1$ |
| | Time constants ($\tau$) | [4, 25] (trainable) |
| | Recurrent connectivity ($p_{\text{rec}}$) | 0.2 |
| | Inhibition probability ($p_{\text{inh}}$) | 0.2 |
| | Dale's principle | applied |
| | Activation function | sigmoid |
| Cue (if applicable) | Number of cues | 2 |
| | Cue onset | 0 or post-stimulus (75) |
| | Cue duration | until trial end |
| | Cue layer index | 3 |
| Training | Batch size | 256 |
| | Learning rate | 0.01 |
| | Iterations (max) | 200,000 |
| | Early stop (loss) | $< 0.001$ |
| | Early stop (accuracy) | $> 0.90$ |

## B  ANALYSIS METHODS

### B.1  DECODING

To evaluate the learned representations of stimulus identities in each ctRNN, we trained least-squares support vector machine (LS-SVM) classifiers to decode stimulus identity based on activity patterns in each hidden layer. For each hidden layer, population activity was averaged within non-overlapping time windows that were 5 time steps wide. The evaluation dataset was generated to be completely balanced (400 trials per stimulus category). The LS-SVM was implemented with `RidgeClassifier` with balanced class weights (from the `scikit-learn` library). Training sets were formed by randomly selecting a random split of 80% of all trials, with the remaining 20% used as a test set. Decoding accuracy was computed for each time window and then averaged across 5 cross-validation splits. We report decoding accuracy averaged across all stimuli, as well as accuracy computed separately for each stimulus.

### B.2  STATISTICS

We tested the effects of bias on model accuracy and decoding accuracy using a repeated-measures ANOVA (AnovaRM, from the `statsmodels` library). To probe within-factor effects, we conducted post-hoc pairwise comparisons across levels within each condition using pairwise_tests (`pingouin`). Reported $P$-values were corrected for multiple comparisons using Bonferroni adjustment.

### B.3  DIMENSIONALITY REDUCTION WITH PCA

To visualize population dynamics over time in the ctRNNs, we performed principal component analysis (PCA) on the neural activity of each layer. For each condition (control: no feedback ablation, FB21: feedback ablation from layer 2 to 1, FB32: feedback ablation from layer 3 to 2), the activity matrix with shape [trials x timepoints x neurons] was reshaped into [trials x timepoints, neurons] to treat each timepoint across all trials as an independent observation. PCA was then applied to reduce the dimensionality of the population activity to the first three principal components. The transformed data were reshaped back into [trials x timepoints x components] for visualization. Separate PCA embeddings were computed for each layer/condition to examine how the population trajectories differed across network layers and experimental conditions.

## C  SUPPLEMENTAL FIGURES

### C.1  FEEDBACK ABLATION

We first explored many levels of feedback ablation. We found that model performance rapidly decreased to floor and remained there for all lower feedback levels. Thus, we selected a decrement of 0.7 to present in the main analysis as it revealed performance deficits but was still higher than the floor for decoding. Figure C.1 includes model performance, decoding accuracy, and bias for ctRNNs trained with (D-F) and without a cue (A-C).

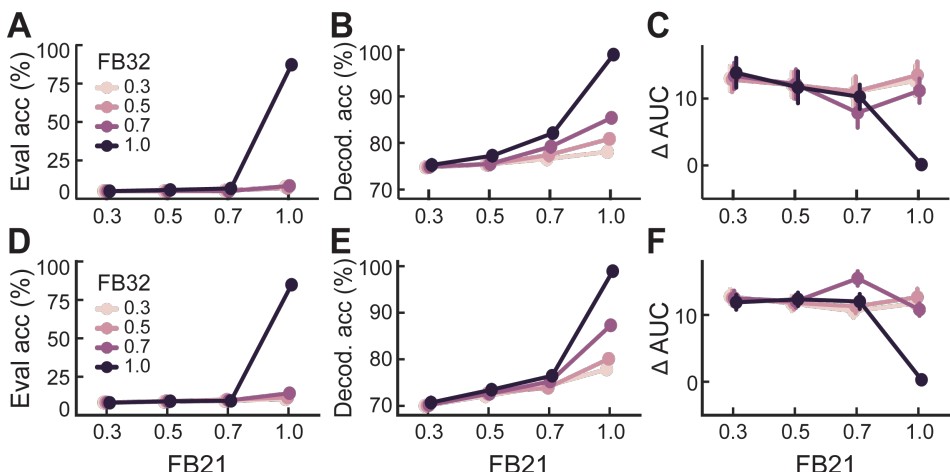

Figure C.1: (A-C) ctRNNs trained without a cue. (D-F) ctRNNS trained with a cue. All error bars are standard error.

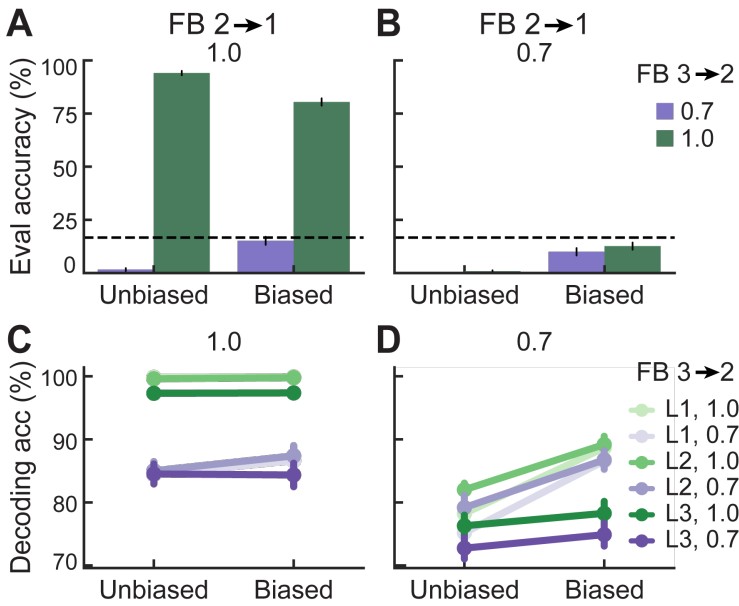

Figure C.2: Analysis of ctRNN performance and representations with feedback ablation. Color represents FB32, green=full, purple=ablated. (A) Evaluation accuracy with full FB21. (B) Evaluation accuracy with reduced FB21 to (C-D)Color saturation represents layer light = layer 1, medium = layer 2, dark = layer 3. (C) Decoding accuracy with full FB21. (D) Decoding accuracy with reduced FB21.

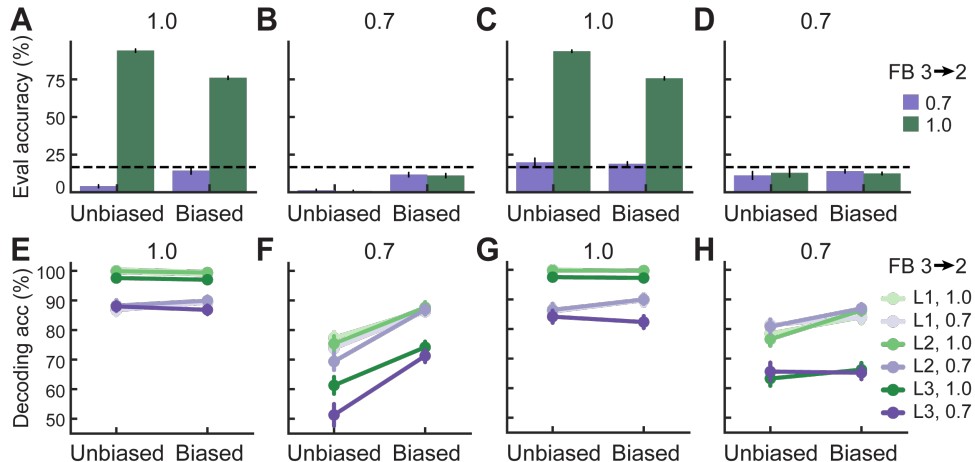

Figure C.3: Analysis of cue ctRNN performance and representations with feedback ablation. Color represents FB32, green=full, purple=ablated. (A) Evaluation accuracy with full FB21 when cue was at the start. (B) Evaluation accuracy with reduced FB21 when cue was at the start. (C) Evaluation accuracy with full FB21 when cue was at the stimulus offset. (D) Evaluation accuracy with reduced FB21 when cue was at the stimulus offset. (E-H) Color saturation represents layer, light = layer 1, medium = layer 2, dark = layer 3. (E) Decoding accuracy with full FB21 when cue was at the start. (F) Decoding accuracy with reduced FB21 when cue was at the start. (G) Decoding accuracy with full FB21 when cue was at the stimulus offset. (H) Decoding accuracy with reduced FB21 when cue was at the stimulus offset.

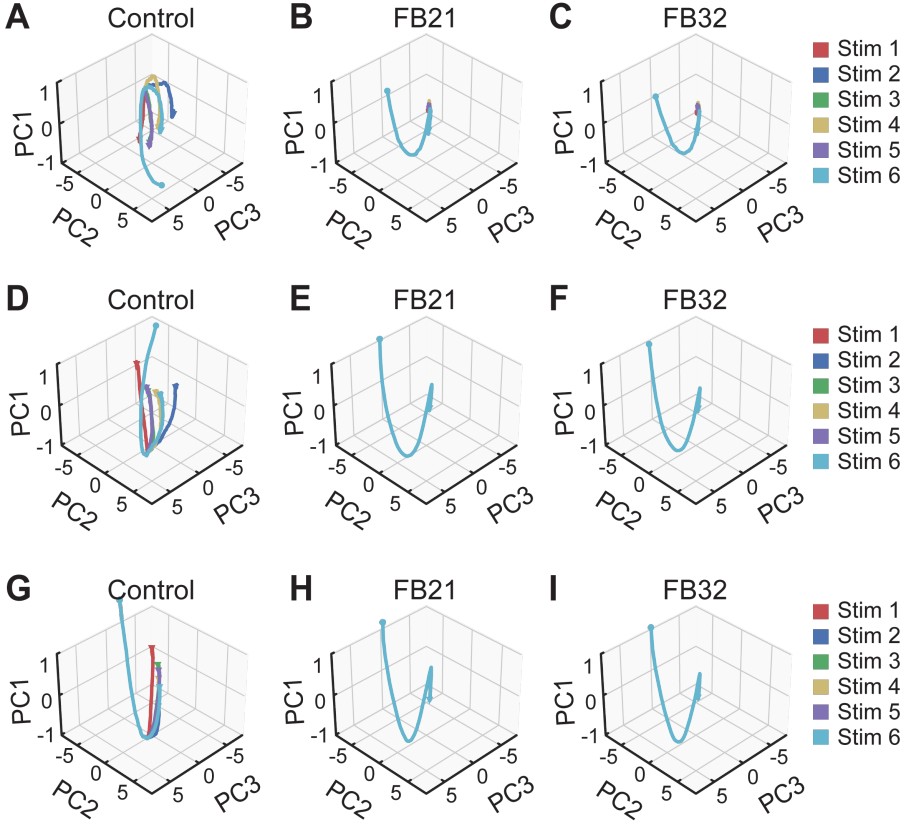

Figure C.4: Example no-cue ctRNN. (A-C) PCA trajectories in example ctRNN layer 1 with (A) full feedback, (B) reduced FB21 only, (C) reduced FB32 only. (D-F) Same for layer 2. (G-I) same for layer 3.

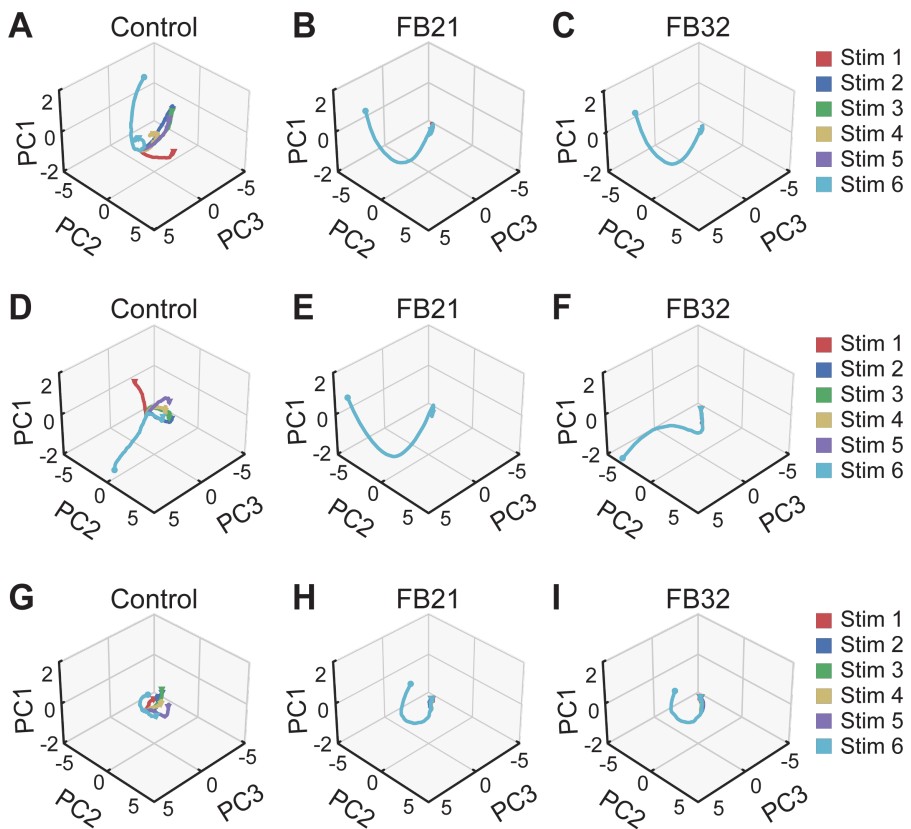

Figure C.5: Example late cued ctRNN. (A-C) PCA trajectories in example ctRNN layer 1 (A) full feedback, (B) reduced FB21 only, (C) reduced FB32 only. (D-F) Same for layer 2. (G-I) same for layer 3.

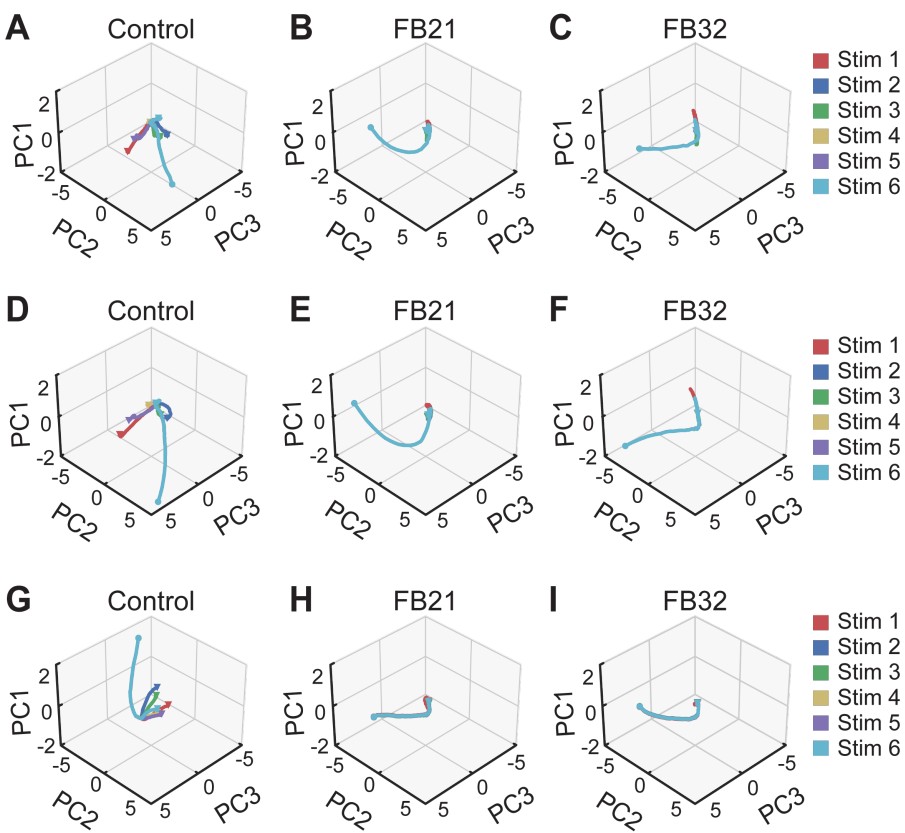

Figure C.6: Example early cued ctRNN. (A-C) PCA trajectories in example ctRNN layer 1 with full feedback(A), ablated feedback from layer 2 to 1 (B), and ablated feedback from layer 3 to 2(C). (D-F) Layer 2. (G-I) Layer 3.

