# OpenReview forum: "Feedback promotes efficient-coding while attenuating bias in recurrent neural networks"
_ICLR.cc/2026/Conference — ICLR 2026 Conference Withdrawn Submission_

### Official Review · Reviewer_FXim · 2025-10-29

**Soundness:** 2
**Presentation:** 2
**Contribution:** 2
**Rating:** 2
**Confidence:** 3

**Summary:**

This study discusses potential roles of feedback connections in recurrent networks. The authors trained a generic recurrent network (RNN) to reconstruct input series. The model has 6 input and 6 output nodes, each of which corresponds to a class. With each class (e.g., class1), one of the input nodes (e.g., input node 1) is activated accordingly, and the model is trained to activate the corresponding output node (e.g., output node 1).

By perturbing feedback connections (from late to early layers), the authors analyzed the changes in hidden activity patterns. Based on their results, they claim that 1) feedback can “facilitate neural processing of high probability stimuli” in recurrent networks and 2) this feedback connection may be useful to create efficient coding in artificial networks such as DNNs.

**Strengths:**

The implications of the results are interesting and may promote interdisciplinary research between neuroscience and deep learning.

**Weaknesses:**

In my opinion, the evidence to both claims is too weak. First, the given task is too simple, and the network model is also too generic to capture neural dynamics associated with high-level cognitive functions. Second, the authors used RNNs, which have feedback connections from late (high-order) and early (low-order) layers, but most DNNs do not have such feedback. Therefore, it seems difficult to justify generalizing the results reported in this study to DNNs.

**Questions:**

1. The authors do not define clearly what biases, expectations, attention and nonuniform stimuli are in this study. These concepts may be relevant to one another, but they are not automatically the same. Detailed explanations of these concepts would improve readability of this manuscript.

2. The model used in this study is a generic recurrent model. Can the authors explain what they mean by “Hierarchical recurrent networks”?

3. Common DNNs are feedforward networks, but the authors are testing RNNs in this study. In DNNs, a feedback loop that can deliver feedback signals in RNNs does not exist. I would like to ask the authors how much of their results in RNNs were generalized into DNNs.

4. In line 205, the manuscript stated  “After training, we evaluated 20 independently initialized models on a new balanced dataset of 2,400 trials.” What does “initialization” mean here? To me, it sounds like the models are randomly constructed, which cannot be true. I presume that some units with stochastic properties (e.g., initial membrane potentials) are initialized randomly, but this should be clearly stated.

5. $\Delta$ AUC is used to measure  the bias in the study, but the authors do not explain why this metric can estimate the bias..

6. What does “hierarchical stimulus-response representations” in line 321 mean? The authors also state, “In order to encourage separate representations of the stimulus and the corresponding response behavior”, which sounds extremely ambiguous. They should clarify what they mean here. The authors may be implying that the earlier layers encode the stimuli, and the late layers encode predictions, but even if this is the case, they do not explain why this is a desirable property of recurrent networks with feedback connections from late to earlier layers.

---

### Official Review · Reviewer_ZsFJ · 2025-10-30

**Soundness:** 4
**Presentation:** 2
**Contribution:** 1
**Rating:** 4
**Confidence:** 4

**Summary:**

This submission investigates the role of feedback connections in shaping representations in recurrent neural networks.
In particular, the authors investigate this effect in the context of training networks to perform a dataset using either balanced or unbalanced training datasets, as well as in the presence or absence of significant input noise.
Experimental results demonstrate:

  - networks trained on a bias stimulus ensemble perform the decoding task better for stimuli sampled more frequently (despite having lower overall performance) when there is significant stimulus noise present.
 - when noise is absent, the internal representations reciprocate the bias in the training dataset even though the output behavior is relatively unbiased.
 - The extent of the performance bias grows across the network hierarchy in a condition where feedback between layers is attenuated.
- These observations are preserved in a task variant that requires a delayed response (triggered by an external stimulus cue rather than at stimulus onset).

**Strengths:**

- Investigating the importance of different architectural features of in terms of the impact on stimulus representations is an interesting research direction.
- The observation that bias emerges/increases in networks when dealing with noisy inputs is both interesting and novel to my knowledge.
- The experiments conducted all seem scientifically sound to me
- The authors are committed to releasing code for the purpose of reproducibility
- The simplified setting allows for meaningful/interpretable ablations that implicate a specific set of parameters/a specific architectural element in a computation/behavioral output (top down feedback in encoding training set induced biases).

**Weaknesses:**

- Experiments are limited to an extremely toy setting, and it is difficult to say whether or not the observations from this paper would generalize to more complex settings (perhaps something like sequential MNIST could be considered)?
- Section 4 seems to offer little to the paper. It seems to me that the only difference this ablation caused was that training networks with late cue timing reduced the impact of down-regulating  feedback in terms of model performance. This did not jump out to my eyes as a particularly obvious or important control, and thus felt slightly orthogonal to the main contributions of this work. I think this sections' contribution would have been more obvious a the main text discussion was limited to where this change had an impact and this result was accompanied by a main text figure.
- The presentation of some results was unclear/hard to parse for me. See below my question about figure 3A for an example.

**Questions:**

- In many figures the task performance of intermediate layers is considered. I assume these were ascertained by training additional linear readouts on the internal layers. Is this the case? If so, was this mentioned in the paper somewhere?
- I am having trouble parsing Figure 3A. What differentiates the colors here? I thought the difference between A and B was whether or not feedback was attenuated but that also seems to be the difference between the colors?
- Typo in lines 207-208: is the noise level 0.1 (0.6) )or 0.01 (0.06)? Main text and figures seem to disagree.
- Even though there is little evidence for long range inhibitory connections, is it not true that feedback signals are/can be routed through local inhibitory interneurons? I am wondering what the impact of this architectural constraint might be.
- What implications does this set of experiments have for our ability to understand computation in more interesting/complex neural systems (be they biological or artificial)? The author's mention this as a future direction, but I think slightly expanding on this by proposing some example experiments could strengthen/make more obvious the contribution of this work.

---

### Official Review · Reviewer_usMX · 2025-10-30

**Soundness:** 2
**Presentation:** 3
**Contribution:** 1
**Rating:** 0
**Confidence:** 4

**Summary:**

This work aims to investigates how artificial neural networks learn from biased or imbalanced data distributions, aiming to reconcile the "efficient-coding" hypothesis from neuroscience with the harmful effects of biased data and poor generalization seen in machine learning. This is investigated by constructing a three-layer continuous time RNN model (ctRNN) which operates to distinguish between a set of six input signals. The authors report that, a ctRNN trained on imbalanced data can perform well and yet when inter-layer feedback connections are manipulated, networks with imbalanced data during training suffer more than models with balanced training schemes. This is taken as evidence that these feedback connections protect the representations of rarer stimuli, such that their ablation causes biases to be magnified and other representations to collapse. This is also tested in models in which a cue is also provided to the output to the network which indicates a change in task (change in mapping) such that top-down information flow could be further confirmed and this effect reproduced.

**Strengths:**

- The overview of existing literature across multiple niches in neuroscience and AI was carried out relatively well and placed this research direction well
- The proposal that feedback connections within biological networks might actively regulate and suppress harmful, prior-induced biases is an interesting and potentially powerful hypothesis.
- Using an intervention-based method (ablation) method was a good start for examining the function of the feedback pathways in this model

**Weaknesses:**

- **Overstated contributions**: The biggest weakness of this work is simply that the decision making setups are so simple as to be unrealistic and potentially uninformative. The simple 1-of-6 reproduction tasks are used to make significant claims of contributing to a better understanding of brains and AI. Quite simply, these extremely limited toy models are a far cry from a real confirmation of the role of feedback signals. Questions are not posed or explored as to whether these results may be a fluke of this dataset or the models constructed. Questions such as: Would this same solution be found for more complex problem sets? Could a particular aspect of the network's initialization or structure be the cause of these observations?
-  **Limited exploration of known issues with AI models**: This weakness links to the above point and expands upon it from an AI and ML perspective. Artificial neural networks are well known for finding solutions which differ depending upon how weights are initialized. For example, to solve the tasks posed in this paper, feedback connections are likely entirely unnecessary. So why is this particular solution found in these networks? I would hypothesize that it is because the feedback connections are initialized rather strongly (or as strongly as feedforward connections) something which is not typically thought to be true in biological neural networks. Furthermore, neural network models are known to suffer from catastrophic forgetting and strong biasing by data biases, effects that are not seen in humans. Questions around whether the effects seen here are purely curiosities of neural network models are also not posed. Instead, it is simply assumed that the same mechanism is likely active in brains.
- **Lack of rigorous alternative testing**: In this work, the feedback connections in the model are perturbed in order to determine whether they are important to overcoming bias in the network's training. However, the other (feedforward and lateral) connections are never tested. Would similar effects not simply show up no matter which connections were ablated? I believe so, considering that this is a fully connected dynamical system in which information circulates. This point is never considered or approach in this setup.

Finally, note that although this manuscript discusses at length the perturbation and manipulation of feedback connections from higher to lower layers, the mathematical description of the model (Equations 1) describe no top-down connectivity, only recurrent and feedforward.

**Questions:**

In the weaknesses section above, I posed a number of issues. Each include questions which I believe should have been answered within this work for completion. Please consider these point by point.

One additional question which remains: Were all models trained to convergence in accuracy? The bias models seem to consistently have lower accuracies than the unbiased models (even in the low noise range). I would expect for such a task that they could both be trained to 100% accuracy, especially in ctRNNs with millions of parameters.

---

### Official Review · Reviewer_iPHi · 2025-10-31

**Soundness:** 2
**Presentation:** 3
**Contribution:** 1
**Rating:** 2
**Confidence:** 5

**Summary:**

The paper uses a multilayer recurrent network with feedback trained to estimate the mean for a set of discrete noisy inputs presented on separate channels one at a time to ask the question of how the presence of feedback affects the quality of processing of uncommon vs frequent inputs. It uses variation in test input noise and modulation of feedback strength to show differential effects when training with a equal across inputs or unequal input frequency  in the training set. Test set is always equal probability. Feedback is shown to influence the biases induced by uneven training statistics, which may have some implications for understanding the role of feedback in representing prior input statistics.

**Strengths:**

The question of the role of feedback in enforcing sensory priors is an interesting one that benefits from modeling and numerical investigation.

**Weaknesses:**

Conceptually the idea of bias in efficient coding is a little weird. The goal of learning  for both efficient coding models in neuroscience and for ANNs trained by optimization, is to minimize error (or other loss) taken in expectation over a given data distribution. Increasing precision of representation for commonly occurring stimuli is normative and not a nuisance or failure. For the brain, out of distribution generalization to unnatural sensory experience is not ecologically relevant so using the ML language of bias and failure to generalize makes little sense in the neuroscience context. That makes part of the premise somewhat suspect. A better way to formulate the scope of the question is to ask what role do local circuit inhomogeneities in coding vs top down feedback effects play in differential processing of common versus uncommon stimuli. But even in that framing, one would need to compare results across multiple tasks for the answer to be compelling.

Learning an identity input output mapping with heavily biased input statistics is very basic in terms of computational complexity. It is not clear what if any of the conclusions generalize to tasks that require some (nonlinear) computation for the input-output map. The top down cue forces feedback to be important by construction so it's not clear to me how much of those results are surprising or educational.

Key elements of the implementation are not described in the main text.

A fraction of the results are trivially expected, e.g. within distribution test accuracy being better than significantly out of distribution.

**Questions:**

Equation 1 lists purely feedforward interaction and none of the feedback connections mentioned in the text and figure. Given the critical nature of these connections for the scientific question it makes it hard to assess the results of numerical simulations. I have operated under the assumption that the reciprocal pattern of connectivity (no skip connections) between layer is what was used but this needs clarification.   This was clarified to be true much later in the results but made methods description confusing, please fix.

What loss function was used for training? L2 ? why not cross-entropy? does it matter?

Small comment: calling a constant mean gaussian iid input as "time varying" is in my opinion misleading and should be corrected

Fig2bc: some lines are not visible? if overlapping please use dashed lines or a small horizontal offset to clear up where the invisible lines lie
are the decoders evaluated on the same (ood for biased training) test distribution?

Fig3b: why are there biases in the network trained with uniform stimulus distribution after feedback ablation? and why is it the largest then the 3 to 2 feedback is left intact (dark green)?

Fig3c: same issues with visuals as for fig 2

Why are the responses so nonstationary given that the task uses completely stationary inputs with iid noise? are the dynamics seen in PC space a reflection propagation time constants? does a layer specific version look more interpretable in terms of the effects of feedback?

What is the dimensionality of the neural activity space, i.e. how much variance does the first 3pcs explain about the data?

Can you understand the nature of the dynamics? e.g. by linearizing dynamics around stimulus specific fixed points?

section 4.2: i fail to understand how training without a cue can possibly resolve the ambiguity between which of the two permutation maps the circuit needs to implement in any given trial. was that blocked somehow or by "no cue" you mean learning a single stationary permutation map?

Similarly it's not clear why performance would not be affected by the late vs early cue distinction: in the late cue you just process the inputs in early layers then implement the map at the last layer?

the text describes the results in figure 4 and 5 but provides essentially no interpretation for any of it. What do the results mean in terms of the nature of representations and the nature of the feedback formed via different task types? can you say more about the interpretation of the results beyond that feedback makes a difference?

can you comment on the importance of recurrence in the context of the tasks presented? not clear that any of the computations require much recurrence except for the purpose of noise averaging, would you expect different effects if the network were feedforward but with additional layer y layer feedback connections making up the full circuit recurrence?

---

### Note · Authors · 2025-11-24

**Comment:**

We thank the reviewers for their feedback. After careful consideration, we have chosen to withdraw the submission so that we can revise the manuscript and submit it to a neuroscience journal that is more appropriate for its scope.

**Withdrawal Confirmation:**

I have read and agree with the venue's withdrawal policy on behalf of myself and my co-authors.